# Feasibility of implementing the World Health Organization case management guideline for possible serious bacterial infection among young infants in Ntcheu district, Malawi

Tanya Guenther[1]*, Gladson Mopiwa[2], Humphreys Nsona[3], Shamim Qazi[4], Regina Makuluni[5], Chancy Banda Fundani[5], Jenda Gomezgani[6], Leslie Mgalula[7], Mike Chisema[5], Salim Sadruddin[8]

1 Unitaid, World Health Organization, Geneva, Switzerland, 2 The Joint TB/HIV Global Fund -ActionAid, Lilongwe, Malawi, 3 IMCI Unit, Ministry of Health, Lilongwe, Malawi, 4 Department of Maternal Newborn Child and Adolescent Health, World Health Organization, Geneva, Switzerland, 5 District Health Office, Ministry of Health, Ntcheu, Malawi, 6 Save the Children Malawi, Lilongwe, Malawi, 7 World Health Organization, Lilongwe, Malawi, 8 Global Malaria Program, World Health Organization, Geneva, Switzerland

* guenther.tanya@gmail.com

## Abstract

### Background

Neonatal sepsis is a leading cause of mortality, yet the recommended inpatient treatment options are inaccessible to most families in low-income settings. In 2015, the World Health Organization released a guideline for outpatient treatment of young infants (0–59 days of age) with possible serious bacterial infection (PSBI) with simplified antibiotic regimens when referral was not feasible. If implemented widely, this guideline could prevent many deaths. Our implementation research evaluated the feasibility and acceptability of implementing the WHO guideline through the existing health system in Malawi.

### Methods

A prospective cohort study was conducted in 12 first-level health facilities in Ntcheu district. Trained health workers identified and treated young infants with PSBI signs with injection gentamicin for 2 days and oral amoxicillin for 7 days, whereas those with only fast breathing were treated with oral amoxicillin for 7 days. Health Surveillance Assistants (HSAs) were trained to promote care-seeking and to conduct home visits on day 3 and 6 to assess infants under treatment, encourage treatment adherence and remind the caregiver to return for facility follow up. Infants receiving outpatient treatment were followed up at health facility on day 4 and 8. The primary outcome was proportion of outpatient cases completing treatment per protocol.

### Findings

A total of 358 infants received outpatient treatment (202 clinical severe infection, 156 only fast breathing) from February to September 2017. Of these, 92.7% (332/358) met criteria for

**Data Availability Statement:** Data are available upon request from the primary author (guenther. tanya@gmail.com)

**Funding:** This study was funded by Global Affairs Canada through a grant to the World Health Organization for the Rapid Access Expansion Program (RAcE). The funder provided salary support for authors [TG, GM, JG, and SS], but did not have any additional role in the study design, data collection and analysis, decision to publish, or preparation of the manuscript. The specific roles of these authors are articulated in the 'author contributions' section.

**Competing interests:** The authors declare that no competing interests exist. Some of the authors are currently and/or were previously employed by not-for-profit organizations including: Save the Children, World Health Organization, and ActionAid. This does not alter our adherence to PLOS ONE policies on sharing data and materials.

treatment completion and 88.8% (318/358) completed the day 4 follow-up. Twelve (3.4%) young infants clinically failed treatment with no reported deaths in those treated at outpatient level. This treatment failure rate was lower than those reported for the simplified regimens tested in the SATT (8–10%) and AFRINEST (5–8%) equivalency trials. More than half of infants (58.1%; 208/358) received HSA follow-up visits on days 3 and 6.

## Conclusion

Study results demonstrate the feasibility of outpatient treatment for sick young infants when referral is not feasible in Malawi, which will inform scale-up in other parts of Malawi and countries with similar health system constraints.

## Introduction

Globally, 2.5 million newborns died in 2017, accounting for nearly half of the estimated 5.4 million under-five deaths the same year [1]. Serious bacterial infection is a leading cause of newborn mortality, contributing to an estimated one in five deaths in the first month of life [1]. In Malawi, 15,000 (41%) of the estimated 37,000 under 5 deaths in 2017 occurred in the first month of life [1]. Most of these deaths were from preventable or treatable causes, namely infections, complications at birth and complications of prematurity [2].

Deaths in newborns with serious infections can be prevented by prompt and effective antibiotic treatment [3]. However, the recommended inpatient treatment options are inaccessible to most families in low-income settings, including Malawi, and represent a major financial cost to the system and to families themselves [3]. In 2015 the World Health Organization (WHO) released a guideline for outpatient treatment of young infants with possible serious bacterial infection (PSBI) with simplified antibiotic regimens when referral is not feasible [3]. Referral is not feasible refers to when families are unable to or unwilling to undertake a referral. Common reasons for refusing referral include distance, cost of travel and treatment, lack of permission from family members, religious and cultural beliefs, concerns around quality of care, poor attitudes of health workers, and lack of child care/other logistical barriers [4–6]. In 2017, WHO and UNICEF released a joint statement recommending that Ministries of Health and partners work together to carefully introduce and implement the WHO guideline to increase access to high quality care for sick young infants at first level facilities [7].

Simplified outpatient treatment for young infants with signs of PSBI delivered at scale when referral is not feasible through a continuum of care approach involving early identification and timely care-seeking from community, diagnosis and treatment at a first level facility, and back-referral to community for follow-up and treatment completion has the potential of preventing many deaths in Malawi. However, before introduction of the WHO guideline, the Government of Malawi required some implementation research to demonstrate the feasibility of delivering simplified antibiotic treatment for young infants with signs of PSBI in outpatient facilities when referral is not feasible. In addition, government decision makers needed to know the extent to which such treatment would be acceptable to families.

## Methods

### Study design and context

This implementation research was prospectively conducted in a cohort of sick young infants. The primary objective was to evaluate the feasibility of treatment of PSBI for young infants up

to 2 months of age delivered through first-level care facilities in Malawi when referral was not feasible. Secondary objectives were to: i) evaluate whether first-level health facilities can deliver quality care for infants with PSBI and fast breathing only, ii) assess the extent to which Health Surveillance Assistants (HSAs) can follow-up young infants with signs of PSBI or fast breathing only in the community, and iii) assess the acceptability by families of such treatment offered at first-level health facilities. We followed the StaRI standards for reporting implementation research studies [8].

The study was conducted from February to September 2017 in first-level health facilities in Ntcheu district, located in central Malawi along the border with Mozambique. In 2016 Ntcheu had an estimated population of 0.59 million [9]. Government health services in the district are provided through three main levels: 1) referral level through the Ntcheu district hospital; 2) outpatient services through 39 first level health centres, including 28 with maternity services; and 3) village level through community health workers called HSAs.

**Dialogue and engagement with policy makers and communities.** At the time of the study, young infants presenting to first-level health facilities with signs of PSBI in Malawi were referred to the district hospitals for inpatient treatment with a combination of injectable gentamicin and penicillin/ampicillin for up to 7–10 days, in line with WHO recommendations [10]. The district hospitals faced challenges to manage sick young infants for the whole district. In addition, families incurred significant costs associated with inpatient treatment such as lodging, purchasing of food, transport and other costs. The Malawian Ministry of Health (MoH) was concerned about the limited access to treatment for sick young infants and, informed by the publication of the WHO guideline for management of PSBI among young infants when referral is not feasible, expressed interest in introducing the new guideline to Save the Children (SC). The MoH, SC US and WHO partnered to introduce the WHO PSBI guideline in Malawi. Several meetings were held at national level to review the guideline and discuss the process of gradual and structured introduction. It was agreed that before introduction of the guideline across the country, implementation research would be conducted to evaluate the feasibility of case management at first-level health facilities when referral is not feasible. Ntcheu was selected by the MoH as the study district. The main reason for selection of Ntcheu was that the Community-Based Maternal Newborn Care (CBMNC) package, updated in 2014 to reflect the *WHO Caring for Newborns at Home* recommendations, was being introduced in this district and the MoH believed that the pregnancy and postnatal visits by HSAs under the revised CMBNC package would complement the implementation research [11–12]. A timeline of the study development and implementation milestones is provided in S1 Table.

The MoH along with technical experts from WHO and SC met with the Ntcheu District Executive Committee and the District Health Management Team (DHMT) to brief them on the WHO guideline and to gauge their interest in conducting the implementation research. The team visited the community and the health facilities to assess current treatment approaches and care seeking behaviour for sick young infants. Following this exercise, the MoH, with support from SC and WHO, developed an implementation research protocol for the "Treatment of Young Infant Infections in Ntcheu" (TYIIN) study. The MoH IMCI unit led the process of adapting the protocol, training materials and data collection tools for Malawi and engaging with district stakeholders. A district-based team was formed to complete these tasks and to support the implementation of the study, including training of health workers and subsequent monitoring and quality assurance. The team comprised 13 members including IMCI master trainers, the district hospital research coordinator, programme coordinators of CBMNC, Safe motherhood, IMCI, and nurses and clinicians working in the maternity, children's ward and the under-five outpatient department (OPD).

Save the Children and the MoH also engaged community structures to raise awareness of the study and seek their inputs on strategies to mobilize communities. A stakeholder meeting was held in Ntcheu District in April 2016 to inform the chiefs from the Traditional Authorities and other non-governmental organizations working in Ntcheu on the study objectives and methods and seek their views on the acceptability of the study by the community members. Community leaders raised concerns regarding the capacity of health facility staff and HSAs to manage the sick young infants and whether families would come back for follow-up and adhere to treatment. In response, participants were assured that comprehensive training would be offered to all health facility staff and HSAs involved in the assessment and management of the sick young infants and that the study team would closely monitor all children receiving treatment. At the end of the meeting, the study was endorsed by all community leaders.

**Study facilities.**   Twelve outpatient facilities and their associated HSAs were selected in October 2015 in collaboration with the district health office (DHO) based on 'best fit' with the following criteria: presence of providers trained in IMCI; catchment population size (aiming for >15,000); proximity to the district centre to facilitate ongoing follow-up and supervision; and mix of government and Christian Health Association of Malawi (CHAM) facilities to reflect the distribution of health facilities in Ntcheu and other districts of Malawi. The sample of 12 facilities included nine government and three CHAM facilities. Facilities managed by CHAM, a large non-governmental provider of health services, are coordinated by the MoH and charge user fees to cover costs of consultation and medical supplies. The MoH has a service level agreement with CHAM to provide maternal and newborn care services free of cost for up to six weeks post-partum, which was extended to up to two months for the study. Government health services are provided free of cost.

Altogether the 12 study facilities served a population of approximately 0.26 million, with an estimated 9,700 annual live births and an estimated 970 cases of PSBI annually (based on 10% incidence) [13–16]. Each facility was staffed with at least one medical assistant/clinical officer and two nurse-midwives and managed community-based services by 10 to 15 HSAs (total of 147 HSAs across study facilities).

**Identification and assessment of sick young infants.**   Sick young infants with clinical signs of PSBI either self-referred by the families or identified in the community by HSAs presented to the study first level health facilities. All infants with PSBI signs presenting to study health facilities were referred to the district hospital. If referral was not feasible, they were classified as only fast breathing in < 7 days old young infants, clinical severe infection (CSI), or critical illness according to the case definitions outlined in the WHO guideline (Table 1) by trained health facility staff [3]. Whereas those young infants 7–59 days old with only fast breathing were managed on outpatient basis without referral. Fig 1 summarizes the approach for identification, referral and management for each category based on WHO guideline [3,17]. Caregivers of infants eligible for outpatient treatment (infants 7–59 days old with only fast breathing, infants 0–6 days old with only fast breathing who refused referral, and infants 0–59 days old with signs of CSI who refused referral) were invited to participate in the study. Health facility staff explained study procedures to caregivers and obtained written informed consent. Caregivers of sick young infants who did not receive outpatient treatment due to reasons of accepting hospital referral or having signs of critical illness were requested to provide written informed consent for outcome assessment of the infant on day 14 through a follow-up visit.

**Outpatient treatment and follow-up procedures.**   Trained facility staff provided outpatient treatment to sick young infants at study health facilities according to the WHO guideline (Table 1). Health facility staff administered the first dose of gentamicin injection and the first dose of dispersible oral amoxicillin on first day of treatment and the second dose of gentamicin

**Table 1. Case definitions and management according to the WHO guideline [3] and study outcomes.**

| Case definitions | Recommendation |
|---|---|
| **Management** | |
| Possible serious bacterial infection (PSBI): young infant with one or more of the following signs: not able to feed since birth or stopped feeding well or not feeding at all, convulsions, severe chest in-drawing, fever (temperature $\geq$ 38˚C), low body temperature (< 35.5˚C), movement only when stimulated or no movement at all, fast breathing (60 breaths per minute or more) in infants 0–59 days of age. | Refer to district hospital except those young infants with only fast breathing in 7–59 days of age |
| Give young infants 7–59 days of age pre-referral treatment (first dose ampicillin 50mg/kg or benzyl penicillin 50,000 units/kg and gentamicin injection 5–7.5mg/kg intramuscularly) as per WHO operational guidelines [6]—except in young infants with only fast breathing. | |
| **Re-categorize if referral not feasible** | |
| ○ Fast breathing only in young infants less than 7 days of age with respiratory rate equal to or greater than 60 breaths per minute and no other signs of illness | Refer to district hospital |
| If referral not feasible: Treat with dispersible oral amoxicillin (50 mg/kg per dose) twice per day for 7 days | |
| ○ Clinical severe infection (CSI): young infant with one or more of the following signs: not feeding well, high body temperature ($\geq$ 38˚C), low body temperature (< 35.5˚C), severe chest in-drawing, movement only when stimulated. | Refer to district hospital |
| If referral not feasible: Treat with once daily gentamicin injection (5–7.5 mg/kg) for 2 days and dispersible oral amoxicillin (50 mg/kg per dose) twice per day for 7 days | |
| ○ Critical illness: young infants with any of the following signs: convulsions, unable to feed at all, no movement on stimulation, unable to cry, bulging fontanelle and cyanosis (SpO2<90%). | Refer to district hospital |
| If referral not feasible: Reinforce referral and provide rescue antibiotic treatment (twice daily ampicillin and once daily gentamicin injection (5–7.5 mg/kg)) | |
| Fast breathing only: young infants 7–59 days of age with respiratory rate equal to or greater than 60 breaths per minute and no other signs of illness | Treat on outpatient basis |
| Treat with dispersible oral amoxicillin (50 mg/kg per dose) twice per day for 7 days | |
| **Outcome definitions** | |
| **Primary** | **Secondary** |
| • Treatment completion: ○ CSI: proportion of young infants with signs of CSI who completed outpatient treatment as per protocol—both doses of gentamicin and at least 12 of 14 amoxicillin doses; ○ Fast breathing only: proportion of young infants with only fast breathing who completed treatment as per protocol—at least 12 out of 14 amoxicillin doses). | • Refusal of referral advice and acceptance of outpatient treatment: proportion of young infants with signs of CSI or only fast breathing 0–6 days of age identified at first-level facilities that accepted outpatient treatment • Mandatory day 4 follow up: proportion of young infants with signs of CSI or only fast breathing treated on an outpatient basis who complete mandatory Day 4 follow-up visit at a health facility • HSA follow up: proportion of young infants with signs of CSI or only fast breathing treated on an outpatient basis who receive a day 3 follow-up home visit from an HSA • Treatment failure: proportion of young infants with signs of CSI or only fast breathing treated on an outpatient basis who experienced clinical deterioration defined as: emergence of any sign of critical illness or a new sign of CSI, persistence of same sign(s) on day 4, not recovered by day 8, re-emergence of any presenting sign after disappearance on day 4 and any severe adverse events* • Outcome of illness: proportion of young infants with signs of critical illness, CSI or only fast breathing who at day 14 were: i) better; ii) still sick; iii) outcome unknown; or iv) dead |

*Severe adverse events: Death; Diarrhoea with severe dehydration; Disseminated and severe rash; Anaphylactic reaction (Within 30 minutes of getting the antibiotic dose, sudden development of breathing difficulty and raised wheals); Stopped passing urine for >12 hours (Renal failure); Cellulitis or abscess at injection site (only in those receiving injections)

injection the following day. Caregivers (usually mothers) were provided the remaining doses of dispersible oral amoxicillin and counselled to give to the infant twice daily until the end of the 7-day treatment period. Health facility staff showed caregivers how to prepare dispersible oral amoxicillin and taught them about what danger signs to look for and what actions to take if they noticed any danger signs.

Outpatient cases were followed up on day 4 and day 8 at the facility and on day 3 and 6 in the community by trained HSAs to reinforce treatment adherence and follow-up. The health facility staff informed the HSAs in their catchment area about PSBI cases within 24 hours through phone calls, short message service (SMS), in-person or by giving the caregiver a form to hand over to the HSA. Caregivers were provided with a follow-up and medicine reminder card on the first day of treatment tailored to case type (CSI or only fast breathing) to encourage adherence to treatment and follow up. Infants who did not return for follow-up on day 4 or day 8 as advised were followed up by TYIIN study staff. Infants with critical illness or who accepted referral to hospital were followed up by TYIIN study staff for outcome assessment at day 14.

**Study training.** Two IMCI master trainers from Malawi received training from WHO facilitators on an updated young infant IMCI module based on the 2015 WHO PSBI guideline [3]. They trained the Ntcheu district facilitators who in turn trained the staff from the 12 study health facilities at the district hospital. Nurses and clinicians responsible for managing sick young infants at the 12 study facilities, who had been previously trained in IMCI, were trained the WHO PSBI guideline, study standard operating procedures (SOPs), data collection tools, consent process and documentation, and caregiver reminder cards. Facility staff were also provided with and trained on the use of pulse oximeters to assess oxygen saturation. Clinical standardization assessment exercises were included in the training.

HSAs and the health facility staff received joint training in CBMNC adopted for Malawi from the draft *WHO Caring for the Newborn at Home* training manual to promote coordination and team work during implementation [11–12]. This training aimed to refresh the HSAs on their role in counselling families on newborn danger signs during pregnancy and postnatal home visits and assessment and referral of sick young infants through CBMNC. The training also oriented the HSAs to the study SOPs and their additional study specific roles to follow-up on treatment adherence and completion on day 3 and 6 for enrolled cases. Facility staff received a refresher training on clinical assessment, including use of pulse oximetry for assessing oxygen saturation.

The study also oriented existing 321 community volunteers, referred to as secret mothers *'amayi achinsinsi'* who were identified by the community as an essential group of women to link the HSAs with the community. Secret mothers are elderly women selected by their communities to serve as a link between women and the health care system, particularly during pregnancy and the postnatal period [18]. The secret mother approach was introduced in 2012 as part of the Safe Motherhood Initiative and their role in linking pregnant women and mothers to HSAs and health facilities had already been quite well established in Ntcheu district at the time of our study [18]. Secret mothers were oriented to encourage mothers to seek routine postnatal services at the health facility, to raise awareness of newborn danger signs and appropriate care seeking and on their role of supporting HSAs in reminding the caregivers whose infants were receiving outpatient treatment to complete their treatment and return to the health facility for follow up.

**Study implementation and monitoring.** The study was implemented for seven months, following an initial two-month pilot phase. Implementation of the study was led by MoH with evaluation and measurement support from the TYIIN study team (see S2 Table for details on study roles and responsibilities). The TYIIN study team comprised four full-time staff: a

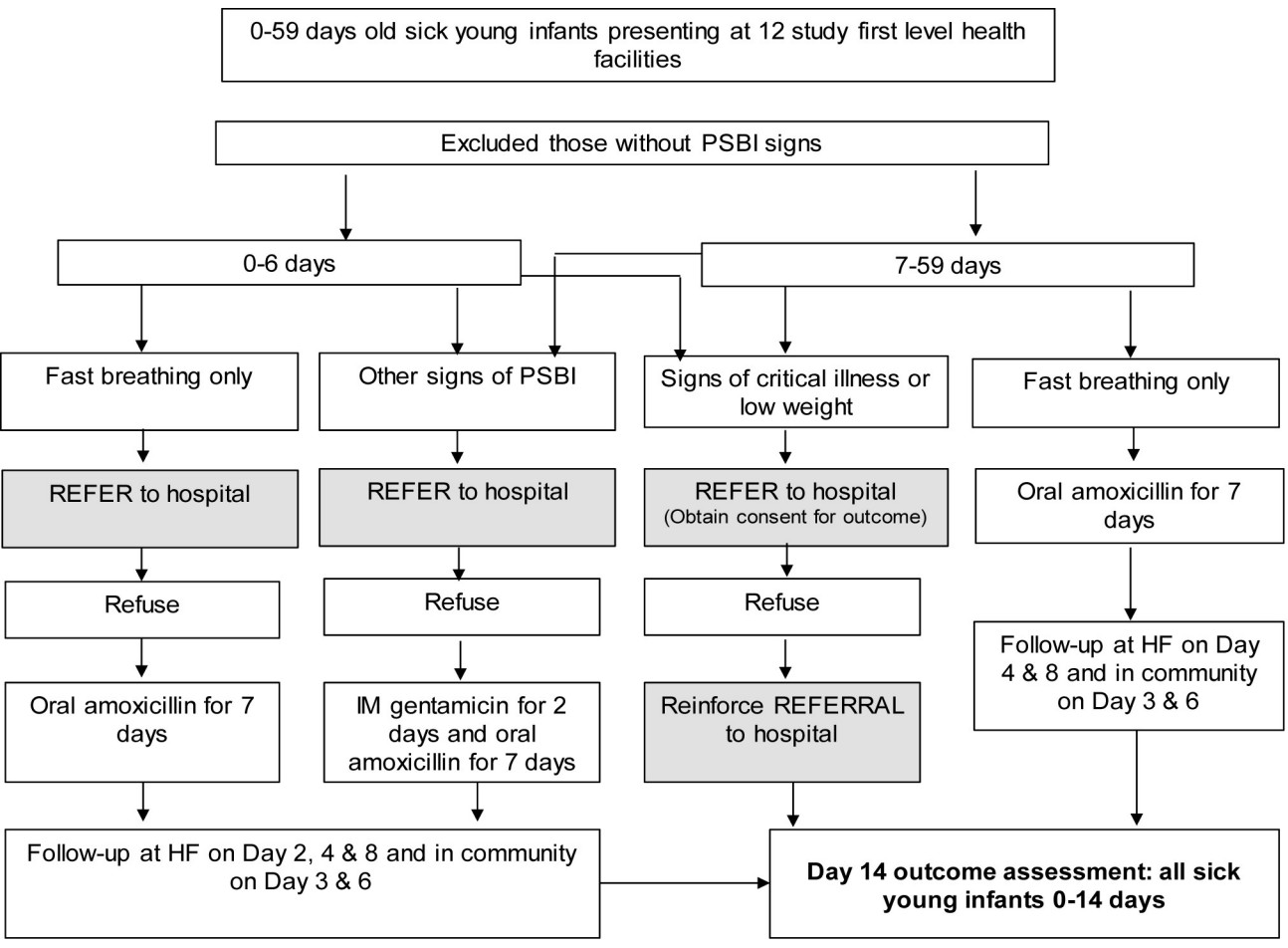

**Fig 1. Case management flow diagram** (refer to definitions in **Table 1**).

manager responsible for overall coordination, two clinical monitors and a data manager. Technical assistance was provided by WHO, SC and the Malawi national IMCI Unit. In addition, the district-based team including members of the Ntcheu DHMT and supervisors from Ntcheu District Hospital supported implementation through trainings, quality monitoring, supportive supervision and mentorship plus any other study-related activities from start to completion.

Prior to initiation of the study, all study facilities were assessed by TYIIN study staff using a facility preparedness checklist to determine availability of essential medicines and equipment and any missing supplies were replenished. During implementation, the TYIIN clinical monitors visited each facility monthly and replenished the supplies, as needed. In addition, facilities were encouraged to notify study staff any time they were approaching minimum stock levels. In case of stock-outs, SC replenished essential study medicines including injection gentamicin, dispersible oral amoxicillin, and injection ampillicin,

Monetary incentives were provided to health facility staff and HSAs involved in the study to compensate for the additional data collection associated with the study. A total of 7,500 kwacha (about $10) per verified outpatient case was given to the health facility in charge who was responsible to share it among all clinical staff and HSA. In addition, the study also provided

5,000 kwacha (about $5) of airtime per facility per month to facilitate communication with study staff and maintain link with HSAs for follow-up.

Study monitoring was conducted by three MoH district staff and the TYIIN study team. Each study facility was assigned to a MoH clinical supervisor for the purposes of study monitoring. These MoH clinical supervisors were responsible to remain in weekly contact with their assigned health facility staff through phone calls and visits where possible. A WhatsApp group was set up for MoH clinical supervisors to update the study team on the enrolment progress of their facilities and discuss study-related issues. The TYIIN clinical monitors and study manager independently supervised study facilities on a monthly basis to verify enrolment, treatment, and assess outcomes. In addition, study review meetings attended by the IMCI Unit, the DHMT, the district study team, health facility nurses and clinicians and HSAs supervisors were held monthly to track progress and address any challenges.

## Outcomes

**Quantitative.** The primary outcome was the proportion of young infants with CSI or only fast breathing aged 0–59 days completing outpatient treatment per protocol. Secondary outcomes included proportion of eligible cases accepting outpatient treatment (refusing referral), completion of day 4 follow up, completion of HSA follow up visits, treatment failure and outcome of the illness at day 14 (Table 1). Health providers checked the adherence to treatment by observing the amoxicillin blister packs on day 4 and 8 facility follow-up visits and by interviewing the child's primary caregiver. The HSAs also tracked treatment adherence during their scheduled contacts with the family on day 3 and day 6 using a similar case recording form. Health provider records were used to determine treatment adherence for study purposes. The TYIIN study clinical monitors and study manager independently validated treatment completion for at least 15% of cases. For cases that failed to return to the health facility for the mandatory day 4 and day 8 follow-up visits, TYIIN clinical monitors and the study manager directly visited the families at their homes to determine treatment adherence and completion. Outcome assessments on day 14 were completed by TYIIN study personnel through home visits or phone calls to families.

**Qualitative.** We also collected qualitative data on feasibility and acceptability through in-depth semi-structured interviews with purposively selected DHO staff, health facility staff, HSAs and mothers of infants receiving outpatient treatment. Interview guides for DHO, health facility staff and HSAs captured experiences and perspectives on successes and challenges in the identification, treatment and follow-up of sick young infants in first level facilities and the community, completing follow-up visits, perceptions of mother's adherence to treatment and follow-up and acceptance of outpatient treatment, and health provider perspectives on whether such an approach should be scaled up in Malawi. Topic guides for mothers of sick young infants addressed decision-making around seeking care, reasons for refusing referral, experiences of outpatient treatment and follow-up, and satisfaction with treatment. Qualitative interviews were conducted by TYIIN study staff and DHMT members in September 2017, towards the end of the study period. The interviews were conducted in Chichewa and detailed notes on responses were recorded in English.

## Sample size and statistical analysis

The sample size for outcome measurement was based on the primary outcome indicator. Our study assumption was that outpatient treatment completion with the simplified antibiotic regimen through first-level facilities in Malawi would be $\geq$ 80%. Considering a significance level of 5%, a power of 80% and a one-sided test, we calculated we would need 368 infants with CSI

or only fast breathing in the study to demonstrate that treatment completion is greater than or equal to 80%.

We conducted the quantitative data analysis using Stata IC 14.2 [StataCorp LP, Texas, USA]. Frequency tables were generated for the primary and secondary outcome variables. We present results separately by case type (CSI or only fast breathing for all variables and critical illness for initial assessment and outcome assessment on day 14). Qualitative data were analyzed using descriptive thematic analysis. Field notes from interviews were reviewed and categorized by the investigators to identify patterns and common themes according to the domains of acceptability and feasibility of outpatient treatment explored during the discussions.

### Ethical considerations

The study received ethical approval by the Malawi National Health Science Research Committee (NHSRC) on September 8, 2015. Written informed consent was obtained from all eligible caregivers prior to their participation in the study. To ensure confidentiality and anonymity during analysis, no personal identifiers were entered into the study database and only study personnel had access to the data.

## Results

### Quantitative findings

A total of 558 young infants 0–59 days (0–6 days: 82; and 7–59 days: 476) were screened at study facilities, of which 378 were identified with signs of PSBI (Table 2). Of these, 156 had only fast breathing, 210 had at least one sign of CSI and 12 had at least one sign of critical illness. Most sick young infants (86.5%; 327/378) were brought directly to the health facility by their caregivers. More than 90% of young infants with signs of CSI or critical illness or fast breathing in 0–6 days old who were referred to the district hospital refused referral (211/226). Thus 358 infants with CSI (n = 202) or only fast breathing (n = 156) received outpatient treatment.

The mean age of the 358 infants receiving outpatient treatment was 26.1 days (standard deviation 17.8); 16.2% were 6 days or younger. There were slightly more males (53.9%) than females. Common presenting signs were: fast-breathing (66.2%); high body temperature (43.9%); and severe chest indrawing (12.6%). Less than 6% of cases presented with low body temperature (2.2%), not feeding well (1.4%), or movement only with stimulation (0.3%). Close to one quarter (23.7%) of infants had two or more signs of illness on the day of assessment.

**Outpatient treatment completion and follow-up.** More than 90% (332/358) of infants met the criteria for treatment completion per protocol and completion rates were similar across case types (Table 3). Most importantly, 95.0% (197/202) of infants with CSI requiring injection gentamicin completed both doses. The mandatory day 4 follow up visit at health facility was completed by 88.8% of cases (318/358) and 80.7% of cases (289/358) completed both day 4 and day 8 facility follow-up visits. More than half (58.1%; 208/358) of outpatient cases received a day 3 and day 6 follow-up home visit by an HSA. Completion of the day 4 facility follow-up visit was significantly higher among infants who received a day 3 follow-up from an HSA (97.2% compared to 89.1%; $p$ = 0.002). Treatment completion rates were similarly high among infants who received a home visit and those who did not (98.1% compared to 95.6%; $p$ = 0.173).

**Outpatient treatment failure.** We documented 12 cases of treatment failure by day 8, representing 3.4% of 358 outpatient cases. These included seven CSI treatment failures (five cases of clinical deterioration and two cases with persistence of same sign(s) on day 4) and five

**Table 2. Results of assessment and screening procedures by PSBI case type (N = 378).**

| | Clinical severe infection (N = 210) | | Fast breathing only (N = 156) | | Critical illness (N = 12) | |
|---|---|---|---|---|---|---|
| | n | % | n | % | n | % |
| **Child age** | | | | | | |
| 0–6 days | 56 | 26.7% | 4 | 2.6% | 2 | 16.7% |
| 7–59 days | 154 | 73.3% | 152 | 97.4% | 10 | 83.3% |
| **Referral outcome*** | | | | | | |
| Accepted referral | 8 | 3.8% | 0 | 0.0% | 7 | 58.3% |
| Refused referral | 202 | 96.2% | 4 | 100% | 5 | 41.7% |
| **Outpatient treatment** | N = 202 | | N = 156 | | NA | |

*-Only infants 0–6 days with fast breathing only were referred as per WHO recommendations (N = 4); Infants 7–59 days with fast breathing were offered outpatient treatment WHO recommendations without referral(N = 152)

only fast breathing cases (three cases of persistence of fast breathing on day 8 and two cases of clinical deterioration).

**Outcome of the illness.** Outcome assessments at day 14 were completed for 95.5% (361/378) of all cases identified, including 150 of 156 cases with only fast breathing, 201 of 210 cases with CSI and 10 of 12 cases with critical illness (Table 4). Among these, all cases who received outpatient treatment had recovered. One death was recorded for a 4-day old infant with critical illness who accepted referral and died on way to hospital.

**Estimated treatment coverage.** We generated an estimate of coverage based on the number of expected PSBI cases. Assuming a 10% incidence of PfSBI for an estimated 5640 lives

**Table 3. Treatment outcomes and follow-up completion by case type among those who received outpatient treatment (n = 358).**

| | Clinical severe infection (N = 202) | | Fast breathing only (N = 156) | | Total outpatient (N = 358) | |
|---|---|---|---|---|---|---|
| | n | % | n | % | n | % |
| **Treatment completion** | | | | | | |
| Completed treatment per protocol* | 187 | 92.6% | 145 | 92.9% | 332 | 92.7% |
| Received both doses of gentamicin | 193 | 95.0% | NA | NA | NA | NA |
| Received one dose of gentamicin** | 5 | 2.5% | NA | NA | NA | NA |
| Received all 14 doses of dispersible amoxicillin | 178 | 88.1% | 135 | 86.5% | 313 | 87.4% |
| Received 10–13 doses of dispersible amoxicillin | 15 | 7.4% | 10 | 6.4% | 25 | 7.0% |
| *Missing information/Lost to follow-up* | 9 | 4.5% | 11 | 7.1% | 20 | 5.6% |
| **Follow-up at health facility** | | | | | | |
| Completed mandatory day 4 follow-up | 179 | 88.6% | 139 | 89.1% | 318 | 88.8% |
| Completed all follow-up visits per protocol (day 4 and day 8) | 166 | 82.2% | 123 | 78.8% | 289 | 80.7% |
| **Follow-up in the community** | | | | | | |
| Received day 3 follow-up | 116 | 57.4% | 96 | 61.5% | 212 | 59.2% |
| Received day 3 and day 6 follow-up per protocol | 114 | 56.4% | 94 | 60.3% | 208 | 58.1% |
| **Treatment failure** | | | | | | |
| Declared as clinical treatment failure¥ | 7 | 3.5% | 5 | 2.2% | 12 | 3.4% |
| *Lost to follow-up* | 7 | 3.5% | 8 | 5.1% | 15 | 4.2% |

*- For clinical severe infection cases: both doses of gentamicin and at least 12 of 14 DT amoxicillin doses; For fast breathing only: at least 12 of 14 DT amoxicillin doses

**Only one child did not receive any doses of gentamicin

¥ -Among 7 clinical severe infection cases: clinical deterioration (5), persistence of same sign(s) on day 4 (2); and (1); Among 5 fast-breathing only cases: persistence of fast breathing on day 8 (3); clinical deterioration (2)

**Table 4.  Outcome of the illness on day 14 for all patients (n = 378).**

| Outcome of the illness on Day 14 | Clinical severe infection (N = 210) | | Fast breathing only (N = 156) | | Critical illness (N = 12) | |
|---|---|---|---|---|---|---|
| | n | % | n | % | n | % |
| Number who were 'better' | 201 | 95.7% | 150 | 96.2% | 9 | 75.0% |
| Number who were 'still sick' | 0 | 0.0% | 0 | 0.0% | 0 | 0.0% |
| Number with outcome unknown | 9 | 4.3% | 6 | 3.8% | 2 | 16.7% |
| Number who died* | 0 | 0.0% | 0 | 0.0% | 1 | 8.3% |

*One infant died on way to district hospital after accepting referral

births during the seven months study period in the population served by the 12 study facilities, we would expect 564 PSBI cases in our study areas. A total of 378 young infants with PSBI were identified and of these, 360 were documented as treated and recovered by day 14, leading to an effective treatment coverage of 63.8%.

## Qualitative findings

In-depth semi-structured interviews were completed with the district IMCI coordinator, four clinical staff from the district hospital pediatric ward (2 pediatric nurses, 1 pediatric clinical officer, 1 in-charge pediatric ward), nine health facility nurses and clinical officers, seven HSAs and seven mothers of sick young infants. The district staff, health facility staff and HSAs were unanimous in their support for the WHO guideline. Clinical staff from the district hospital noted reduced congestion in the paediatric ward after implementation of the guideline: *"There is reduced admission of neonates in the ward and congestion has been reduced."* [Nurse In-charge of paediatric ward]. District hospital staff and the IMCI coordinator also expressed confidence in the ability of health providers at first level facilities to manage cases and in caregivers to complete follow-up, with the IMCI coordinator stating: *"The initial assessment is being done well and they (health facility staff) are able to manage them correctly. The study has shown that hospital management is not the only option. These children can be treated at home with proper follow-up and caregivers are able to follow treatment at home."*

Nurses and clinical officers at the health facilities and HSAs indicated that outpatient treatment was well accepted by families, citing benefits including reduced travel time and costs associated with treatment and improved adherence to the treatment and follow-up protocols, leading to better outcomes for vulnerable babies:

**Table 5.  Main challenges associated with implementation of the WHO guideline and potential solutions mentioned by health facility staff and HSAs.**

| Challenges | Solutions |
|---|---|
| Poor referral systems, particularly lack of ambulances for transport of critical cases | DHO to provide fuel/support for referral transport and to improve referral notification and feedback to health facilities |
| Communication issues between facility staff and HSAs due to weak mobile network, limited airtime and busy schedules | Continuous supervision of health facility staff and HSAs through join review meetings and refresher trainings to reinforce regular communication; provision of airtime to offset mobile costs. |
| Large distances for HSAs to cover for follow-up visits, particularly for non-resident HSAs, and challenges locating families especially those residing outside the facility catchment area | Consider reducing follow-up visits for HSAs and focus on ensuring mandatory facility follow-up completed at first level facility. Enhance support to HSAs with the most difficult catchment areas to address transport issues and motivate them (provide bicycles, etc). |

*"The babies were recovering and families were happy to receive care at home and the treatment was well adhered to"* [Nurse/midwife]

*"Most mothers have fears of the district hospital as they think it's where many babies die. They were happy as they won't spend as much money as they will be at home."* [HSA]

*"In the past caregivers were told that if you see your infant is sick you must go to the district hospital. This made communities keep young infants (at home) in fear of going to the district hospital. But now the community is happy and more patients are seen here now."* [Nurse midwife]

Similarly, all seven mothers interviewed were highly supportive of outpatient treatment, mentioning the reduced costs associated with treatment, better conditions and quality of care they received compared to district level: *"There are a lot of people in the children's ward, up to four babies for one bed. But this treatment at the health centre–we're getting treatment but from home. That money can be used to buy other things at home"* [Mother]. When asked about adherence, all mothers reported it was easy to follow the treatment advice and that they had received detailed information from health staff on how to administer the amoxicillin and when to return for follow-up. They appreciated the follow-up visits at facilities, with one mother reporting *"The nurses were able to explain again how to use the medicines and the baby was being checked"*.

## Implementation challenges and their solutions

Health facility staff and HSAs mentioned several challenges they had experienced with implementation and proposed solutions. The most common challenge reported was poor referral systems, and specifically the lack of transport (Table 5). Several facility staff and HSAs mentioned lack of transport for HSAs to travel around their large catchment areas, especially those residing outside their assigned catchment areas. Other challenges with HSA follow-up included locating families for follow-up–particularly those that came from neighbouring country Mozambique, which bordered Ntcheu district or who lacked a specific follow-up address or had an incomplete family name. When asked about what challenges they anticipated if the approach was expanded, most facility staff mentioned concerns around stock-outs of medicines, issues of airtime and network connectively to liaise with district staff for referral of critical cases and for reaching HSAs to do follow-up. Facility staff recommended ongoing supervision and continuous capacity building sessions, including through review meetings, for HSAs and facility health workers on the WHO guideline to reinforce clinical skills and communication pathways.

## Discussion

Our findings indicate that appropriate implementation of the latest WHO PSBI management guideline for sick young infants is feasible at first-level facilities in Malawi where referral to a hospital is not possible. In nearly all cases, families refused referral to a hospital and accepted outpatient treatment, showing the need for provision and uptake of care at this level. We documented high levels of caregiver adherence with outpatient treatment and facility follow-up, both of which are essential to ensure quality outcomes for vulnerable infants. The levels of treatment adherence and follow-up in our study were similar to those reported by other WHO PSBI guideline implementation research study sites [unpublished data, personal communication] and treatment failure rates were lower than those reported in the SATT (8–10%) and AFRINEST trials (5–8%) [13–16].

While our study was not designed to measure treatment coverage, we were able to generate an estimate based on the number of expected cases and the number of cases documented as receiving treatment and recovered. Our estimated effective treatment coverage level of 63.8% is similar to those reported by a few of the other WHO PSBI guideline implementation research sites [unpublished data, personal communication]. In most instances, families sought care directly from health facilities, with less than 15% of cases being identified and referred by HSAs to the first level health facilities. This was not unexpected, given that HSAs are trained to conduct home visits during the first week of the postnatal period and that coverage levels of home visits are low, even with substantial program support [19–20]. However, an estimated one-third of expected cases were missed, particularly very young infants less than 7 days old, which suggests there is a problem identifying and bringing young infants for care in the vulnerable first week of life. Further efforts to identify sick newborns and sensitize communities around newborn illness and address barriers to appropriate care-seeking are needed to encourage uptake of the service by the caregivers and the community. Levels of facility delivery in Malawi exceed 90% [21], providing the opportunity to strengthen pre-discharge assessment of newborns and counselling of mothers and other caregivers on newborn danger signs and care-seeking. At the community level, village health committees, HSAs, and secret mothers (where available) could be supported to empower mothers to identify sick young infants and raise awareness of outpatient treatment services and facilitate referrals for families as needed.

We found less than optimal levels of follow-up by community-based HSAs, even with the support of small monetary incentives. In Ntcheu, and most districts across Malawi, many HSAs live outside their designated areas of work and may be in their catchment area only two days out of a week, spending the remainder of their time in the health facility and leaving little time for conducting follow-up visits. In addition, interviews with HSAs and providers identified challenges with locating families, long distances for HSAs living outside their catchment area, and competing priorities given the multi-task nature of HSA's work. Our results showed that infants who received an HSA home visit on day 3 were significantly more likely to return to the facility for the day 4 follow-up, while treatment completion rates were high regardless of HSA follow-up. Given the potential value of involving HSAs to encourage families to attend the facility for follow-up, approaches to optimize HSA follow-up under scale-up conditions should be explored as implementation of the WHO guideline is expanded in Malawi. There may also be scope to further explore the extent to which secret mothers could be engaged to support HSAs and connect them with families of infants under treatment, which this study did not directly measure.

The main strength of our study was the central role played by the MoH in the study design, implementation, analysis and dissemination of the results. This strong support was essential for generating ownership and buy-in at the district level and promoting quality assurance of study implementation. The TYIIN study team worked closely with the district staff to supervise all aspects of the study and mentor facility staff, which contributed to the quality of implementation through existing systems and staff. The MoH covered the majority of costs associated with treatment, with the exception of ampicillin, which was not on the essential drug list for first level facilities at the time of the study.

There were several limitations to our study. While outcome assessments were completed for 95% of sick young infants, the 5% for whom outcome assessments were missed were largely the most critical cases and those who accepted referral. Due to limited study resources, the TYIIN study team focussed on completing outcome assessments for outpatient cases. As a result, it is possible that the mortality rate among PSBI cases could have been higher, particularly for those with critical illness. The study was implemented for only seven months and we were not able to assess the extent to which the quality of implementation would be sustained

over time and under normal programmatic conditions. Small incentives were provided to health facility staff and HSAs to offset the additional burden of recording for study purposes, which may have generated more motivation to manage sick young infants than would be present under scale-up conditions in the absence of incentives. However, qualitative interviews with health facility staff revealed high levels of satisfaction with the WHO guideline and the opportunity to provide quality treatment closer to home that suggests broad acceptance of the approach that will sustain over time. We conducted the study in one district, in which the community-based package to promote home visits by HSAs during pregnancy and the postnatal period was being strengthened concurrently, and this may have biased the study towards greater care-seeking and higher levels of follow-up. While the generalizability of the findings to other parts of Malawi is unknown, the revised CBMNC package has now been rolled out in most districts. In addition, our study included a mix of government and CHAM facilities that reflects the national distribution of first level facilities in Malawi. Our study did not find any notable differences in treatment completion and follow-up levels between CHAM and government facilities, but the sample size was small and for the study CHAM facilities extended free treatment from 0–6 weeks to cover the full young infant period (~8 weeks); this is an area for further investigation as outpatient treatment of PSBI when referral is not possible expands in Malawi. Finally, the qualitative component of our study involved a small sample of purposively selected respondents and detailed field notes rather than full transcriptions were used for analysis. However, despite the small sample, we found saturation in responses for our main lines of inquiry.

Since the dissemination of the study findings in late 2017, the MoH has taken steps to scale up access to outpatient treatment of PSBI in young infants, including changing the IMCI policy and adapting material for scale-up, including a flow chart on the WHO guideline and developing costed plans for training and medicine supplies [personal communication, Malawi Ministry of Health]. As of June 2019, the MoH with support from partners has expanded the approach across all health facilities and their respective HSAs in Ntcheu district and introduced the WHO PSBI guideline in an additional 11 of Malawi's 28 districts at first level facilities, training at least one health worker per facility (out of 2 or 3 eligible providers). The MoH reports that while trained providers are able to follow the WHO guideline, there are gaps in case management when the trained provider is not on shift and with maintaining adequate stocks of dispersible amoxicillin and medical supplies such as thermometers and timers. The lessons learned during implementation have important implications for the continuing scale up of outpatient treatment for sick young infants in Malawi and beyond:

- Community sensitization to increase care-seeking for sick young infants is essential–starting with local leaders and families. Standardized operating procedures for community groups would help optimize engagement in health service delivery at both community and facility level.

- Training sessions that bring together HSAs, health facility nurses and clinicians, strengthened working relationship and management of sick young infants. Ongoing supervision and regular capacity building sessions for HSAs and facility health workers on sick young infant guidelines will help ensure adherence in routine practice and maintain skills.

- While many HSAs residing in their catchment area were able to conduct home visits, different strategies may be needed for those who live outside their catchment area (which can be a substantial proportion in some areas of Malawi, particularly in more remote villages).

## Conclusion

The study has demonstrated the feasibility of a scalable model for improving the access to management of PSBI and only fast breathing in young infants at first-level facilities in Malawi when hospital referral is not feasible. The leadership of the MoH in the design, implementation and dissemination of the study findings facilitated further scale-up of the WHO guideline. Ongoing support and oversight are needed to ensure that quality of implementation is maintained and that sick young infants in Malawi access optimal clinical care.

## Supporting information

**S1 Data.**
(XLSX)

**S1 Table. Timeline of TYIIN study milestones.**
(DOCX)

**S2 Table. Summary of roles and responsibilities for TYIIN study.**
(DOCX)

## Acknowledgments

We are grateful to the health facility staff, HSAs, secret mothers and to the caregivers who participated in the study. We acknowledge the support for the study provided by Save the Children Canada, Save the Children Malawi, and Save the Children US. We thank Yasir Bin Nisar and Samira Aboubaker for their critical review and inputs to the paper.

## Author Contributions

**Conceptualization:** Tanya Guenther, Humphreys Nsona, Leslie Mgalula, Salim Sadruddin.

**Formal analysis:** Tanya Guenther, Gladson Mopiwa, Shamim Qazi, Regina Makuluni, Salim Sadruddin.

**Funding acquisition:** Tanya Guenther, Humphreys Nsona, Salim Sadruddin.

**Investigation:** Gladson Mopiwa, Humphreys Nsona, Shamim Qazi, Salim Sadruddin.

**Methodology:** Tanya Guenther, Gladson Mopiwa, Humphreys Nsona, Shamim Qazi, Jenda Gomezgani, Mike Chisema, Salim Sadruddin.

**Project administration:** Gladson Mopiwa, Humphreys Nsona, Regina Makuluni, Chancy Banda Fundani, Jenda Gomezgani, Leslie Mgalula, Mike Chisema, Salim Sadruddin.

**Supervision:** Gladson Mopiwa, Humphreys Nsona, Shamim Qazi, Regina Makuluni, Chancy Banda Fundani, Jenda Gomezgani, Mike Chisema, Salim Sadruddin.

**Writing – original draft:** Tanya Guenther, Salim Sadruddin.

**Writing – review & editing:** Tanya Guenther, Gladson Mopiwa, Humphreys Nsona, Shamim Qazi, Regina Makuluni, Chancy Banda Fundani, Jenda Gomezgani, Leslie Mgalula, Mike Chisema, Salim Sadruddin.

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
