## [Decision Letter · Decision Letter 0]

19 Nov 2019

PONE-D-19-25252

Feasibility of implementing the World Health Organization case management guideline for possible serious bacterial infection among young infants in Ntcheu district, Malawi

PLOS ONE

Dear Ms. Guenther,

Thank you for submitting your manuscript to PLOS ONE. After careful consideration, we feel that it has merit but does not fully meet PLOS ONE’s publication criteria as it currently stands. Therefore, we invite you to submit a revised version of the manuscript that addresses the points raised during the review process.

We would appreciate receiving your revised manuscript by Jan 02 2020 11:59PM. To enhance the reproducibility of your results, we recommend that if applicable you deposit your laboratory protocols in protocols.io, where a protocol can be assigned its own identifier (DOI) such that it can be cited independently in the future. For instructions see: http://journals.plos.org/plosone/s/submission-guidelines#loc-laboratory-protocols

We look forward to receiving your revised manuscript.

Kind regards,

Emma Sacks

Academic Editor

PLOS ONE

Journal Requirements:

1. Thank you for including your competing interests statement;

"The authors have declared no competing interests exist."

We note that one or more of the authors are employed by a commercial company: Save the Children Malawi, World Health Organization, Ministry of Health, The Joint TB/HIV Global Fund -ActionAid, and Partnership for Human Development -ABT Associates.

Additional Editor Comments:

This is a well done study about an important implementation topic. As an implementation study, the detail about the process of introducing the study is appreciated, as are the efforts to understand the context and challenges with uptake of a guideline.

The most important issue to address is that as Ntcheu was selected because of ongoing complementary programs, the authors must address the potential for BIAS, as Ntcheu may be quite different than other sites and uptake potentially much higher. Further, the implementation of district based teams, quality assurance efforts, and extension of the user-fee waiver, sounds very intensive, and the authors should address the feasibility that this could be replicated in a program setting.

In addition to the review comments, can you please also address the following:

-In the abstract, can you please include if the finding that 3.4% of young infants clinically failing treatment is similar or different to the PSBI efficacy trial?

-The authors state that sepsis treatment in children is a large cost to the financial system; shouldn't this fall under the most basic primary care and not be subject to cost efficiency measures?

-Acceptability and uptake are not the same; the former should be measured through qualitative methods

-please define terms such as "ease of access" and "referral not feasible"

-Please add page numbers

Reviewers' comments:

Reviewer's Responses to Questions

**Comments to the Author**

1. Is the manuscript technically sound, and do the data support the conclusions?

Reviewer #1: Yes

Reviewer #2: Yes

2. Has the statistical analysis been performed appropriately and rigorously? 

Reviewer #1: Yes

Reviewer #2: Yes

3. Have the authors made all data underlying the findings in their manuscript fully available?

Reviewer #1: Yes

Reviewer #2: Yes

4. Is the manuscript presented in an intelligible fashion and written in standard English?

Reviewer #1: Yes

Reviewer #2: Yes

5. Review Comments to the Author

Reviewer #1: This is a well written clear manuscript. The only section that could do with additional explanation is the 'secret mothers'. please add a few more sentences regarding their role within the context of PSBI. Are they widely used, already in existence?

how do you know that they are effective?

Reviewer #2: Dear Editors,

Thank you for inviting me to review this manuscript. it was my privilege to do so.

Overall, this is a superbly written manuscript. It is thorough and thoughtful, and the conclusions reached and next steps pragmatic and not-overreaching. A few minor thoughts/suggestions/questions.

1. It would be interesting to understand the reasons given by the parents who refused referral, in this setting. While there is some literature around this, for Malawi, and in particular for Ntcheu, it would be illustrative to understand these reasons. Such reasons may focus future policy and programmatic efforts.

2. Of the nine government and three CHAM facilities, were the results and follow-up similar? While statistical analysis may be precluded here due to small sizes, not infrequently there are differences in health care provision between CHAM and government facilities, and it will be interesting to examine whether follow-up and outcomes are comparable across both settings.

3. The incentives given to health facilities and HSA appeared to preclude the "secret mothers". How much of a role did the secret mothers play in this program? Might follow-up rates have been higher with their active engagement? Or incentivization?4. While it was good to understand in absolute terms how much these incentives were, it would be helpful to the reader to have from a health systems/health economics standpoint some reference points so as to understand whether these incentives are truly sustainable by the Ministry of Health. For e.g. how much money is spent on health per person in that district?

5. Given what seems like a successful program, is there a line of sight to line-item budget to continue or expand on this program? Has there been policy change yet?

6. Qualitative interviews - how were these chosen? Were they purposive? Was saturation reached with responses? More detailed methods in this section would make this well-written paper even more robust.

6. PLOS authors have the option to publish the peer review history of their article (what does this mean?). If published, this will include your full peer review and any attached files.

Reviewer #1: Yes: Charlotte E Warren

Reviewer #2: Yes: Cyril M Engmann, PATH & University of Washington

---

## [Author Response · Author response to Decision Letter 0]

8 Jan 2020

Response to reviewers:

Part 1: Editor comments and responses

1. Thank you for including your competing interests statement;

"The authors have declared no competing interests exist."

We note that one or more of the authors are employed by a commercial company: Save the Children Malawi, World Health Organization, Ministry of Health, The Joint TB/HIV Global Fund -ActionAid, and Partnership for Human Development -ABT Associates.

Response: Regarding the revised competing interest statement – we would like to clarify that Save the Children and ActionAid are not commercial companies– rather not-for-profit organizations; World Health Organization (WHO) is a UN affiliate organization; and Malawi Ministry of Health (MoH) is a government entity. We have included in the statement that some authors worked for these entities as requested, while also indicating that no competing interests exist.

We have amended the funding statement and competing interests statement in the cover letter as requested. Please note that since this paper was submitted I have changed positions and no longer work with Abt Associates in Timor-Leste. I currently work for Unitaid at the WHO and have updated my affiliation accordingly. Whereas at Save the Children I received small part of my salary for working on the paper as indicated in the revised funding statement, while at Abt Associates and Unitaid I worked on the paper in my personal time and received no further compensation. 

 Additional Editor Comments:

This is a well done study about an important implementation topic. As an implementation study, the detail about the process of introducing the study is appreciated, as are the efforts to understand the context and challenges with uptake of a guideline.

The most important issue to address is that as Ntcheu was selected because of ongoing complementary programs, the authors must address the potential for BIAS, as Ntcheu may be quite different than other sites and uptake potentially much higher. Further, the implementation of district based teams, quality assurance efforts, and extension of the user-fee waiver, sounds very intensive, and the authors should address the feasibility that this could be replicated in a program setting.

Response: Thank you for raising this important point. The Community Based Maternal and Newborn Care (CBMNC) program was developed in Malawi in 2007 and many Health Surveillance Assistants (HSAs) across the country had been trained in the package since that time, although coverage of the pregnancy and postnatal home visits was low. In 2014, WHO and the MoH collaborated to update the CBMNC package to reflect the WHO recommendations on pregnancy and post-natal home visits and introduced the revised version in Ntcheu district, one of the 8 districts funded through the RAcE program – we have clarified this in the methods section (TC version, page 6, lines 127-131) and also modified the discussion section accordingly (TC version, page 24, lines 514-519). Regarding feasibility under normal program conditions, we have also further expanded the discussion section to address some of these issues (TC version, page 24, lines 508-525)

In addition to the review comments, can you please also address the following:

-In the abstract, can you please include if the finding that 3.4% of young infants clinically failing treatment is similar or different to the PSBI efficacy trial? 

Response: Abstract revised to indicate that the treatment failure rate was less than those reported for the simplified regimens tested in the SATT (8-10%) and AFRINEST trials (5-8%) (TC version, page 2, lines 55-57). We also included reference to this in the discussion section (TC version, page 22, lines 455-457).

-The authors state that sepsis treatment in children is a large cost to the financial system; shouldn't this fall under the most basic primary care and not be subject to cost efficiency measures? 

Response: Hospital treatment of sepsis, particularly complicated cases, can be costly as cases are admitted for inpatient treatment for multiple days. Hospital treatment is also expensive for most families, who may live far away and incur additional costs related to the treatment, such as transport, lodging, food, etc. Identifying cases earlier and managing them at outpatient level has potential to save costs for government-funded health systems and for families. 

-Acceptability and uptake are not the same; the former should be measured through qualitative methods 

Response: We agree that acceptability and uptake are not the same. As part of the qualitative component of the study we interviewed a small sample of mothers whose children had received outpatient treatment to better understand their perceptions of the treatment. We have modified the discussion to better distinguish (TC version, page 21, line 51)

-please define terms such as "ease of access" and "referral not feasible" 

Referral not feasible refers to when families are unable to or unwilling to undertake a referral and we have added this definition to the introduction (TC version, page 4, lines 76-80). Ease of access referred to the proximity of health facilities to facilitate follow-up – we have revised the text in the methods section to clarify (TC version, page 7, line 165).

-Please add page numbers – 

Response: added page numbers.

Part 2: Reviewers comments:

Reviewer's Responses to Questions

Comments to the Author

1. Is the manuscript technically sound, and do the data support the conclusions?

Reviewer #1: Yes

Reviewer #2: Yes

 2. Has the statistical analysis been performed appropriately and rigorously? 

Reviewer #1: Yes

Reviewer #2: Yes

 3. Have the authors made all data underlying the findings in their manuscript fully available?

Reviewer #1: Yes

Reviewer #2: Yes

 4. Is the manuscript presented in an intelligible fashion and written in standard English?

Reviewer #1: Yes

Reviewer #2: Yes

5. Review Comments to the Author

Reviewer #1: This is a well written clear manuscript. The only section that could do with additional explanation is the 'secret mothers'. please add a few more sentences regarding their role within the context of PSBI. Are they widely used, already in existence? how do you know that they are effective?

Response: Thank you for your suggestion. Secret mothers are older women selected by their communities to serve as a link between women and the health care system, particularly during pregnancy and the postnatal period. The concept of secret mothers was started by the MoH in 2012 as part of the Safe Motherhood Initiative and secret mothers are operational in 13 districts of Malawi, including Ntcheu. Secret mothers were already been quite well established in Ntcheu district at the time of the PSBI study in 2017 and we were able to train existing secret mothers. However, to our knowledge their effectiveness has not been studied (we could find no published studies documenting this). We have included additional information on the role of secret mothers in the methods section and also a citation describing secret mothers to direct readers for further details (TC version, page 11, lines 240-245).

Reviewer #2: Dear Editors,

Thank you for inviting me to review this manuscript. it was my privilege to do so.

Overall, this is a superbly written manuscript. It is thorough and thoughtful, and the conclusions reached and next steps pragmatic and not-overreaching. A few minor thoughts/suggestions/questions.

1. It would be interesting to understand the reasons given by the parents who refused referral, in this setting. While there is some literature around this, for Malawi, and in particular for Ntcheu, it would be illustrative to understand these reasons. Such reasons may focus future policy and programmatic efforts.

Response: Thank you for this comment – as part of the qualitative component we asked a small number of mothers why they refused referral and the most common reasons cited were lack of transport to the referral hospital, the need to look after other children and the high costs associated with receiving inpatient treatment at the hospital. We also asked health providers why families refused referral. They also cited the same underlying issues (transport, financial costs, other responsibilities and preference to be treated closer to home). We have included some of the common reasons for refusal of referral in the introduction (TC version, page 4, lines 76-80).

2. Of the nine government and three CHAM facilities, were the results and follow-up similar? While statistical analysis may be precluded here due to small sizes, not infrequently there are differences in health care provision between CHAM and government facilities, and it will be interesting to examine whether follow-up and outcomes are comparable across both settings.

Response: Thank you for this – as noted the sample size was quite small – we have reviewed the treatment completion and follow-up rates between CHAM and government health facilities and the levels were similarly high in both types of facilities. For the research, participating CHAM facilities expanded free treatment to infants from up to six weeks to up to 8 weeks (aged 0-59 days). This would be an important area to track though as implementation expands outside of a research setting and we have noted this in the discussion (TC version, page 24, lines 521-525).

3. The incentives given to health facilities and HSA appeared to preclude the "secret mothers". How much of a role did the secret mothers play in this program? Might follow-up rates have been higher with their active engagement? Or incentivization?

Response: Incentives were only given to the formal providers in the system and primarily to facilitate communication between facility providers, HSAs and study staff (air time) and offset the additional burden of data collection due to the study forms that would not be present under normal conditions. Secret mothers were not required to collect any data. Based on information provided during the qualitative interviews, the contribution of secret mothers to the program was mixed, with some HSAs reporting they were helpful in linking them with families, while others indicated they were able to connect directly without secret mothers. As such, it is not clear whether follow-up rates could have been higher with incentivization of secret mothers. However, scale up of the program beyond the research will not involve incentives at any level of the system and that some districts will not have secret mothers in place. The possible role of secret mothers could be an area for further exploration and we have noted this is the discussion (TC version, page 23 lines 490-492).

4. While it was good to understand in absolute terms how much these incentives were, it would be helpful to the reader to have from a health systems/health economics standpoint some reference points so as to understand whether these incentives are truly sustainable by the Ministry of Health. For e.g. how much money is spent on health per person in that district?

Response: As noted above, monetary incentives were provided to health facility staff and HSAs only for study purposes to offset the additional communication costs and data collection burden. It is not anticipated that the MoH will continue to provide these incentives – we have noted this in the limitations (TC version, page 24, lines 510-511).

5. Given what seems like a successful program, is there a line of sight to line-item budget to continue or expand on this program? Has there been policy change yet?

Response: As mentioned in the discussion, the MoH has started to expand this approach and has developed costed plans for the roll-out and worked with funding and implementation partners to complete the roll-out in Ntcheu district and train providers in 11 additional districts. The MoH reports policy change on PSBI to the IMCI approach policy, and the MoH has updated guidelines for the management of sick young infants and children. The IMCI chart booklet was also updated from the generic WHO guidelines on cough, fast breathing and chest in-drawing pneumonia. Further, the Government has committed funds towards PSBI roll out to cover training and orientation of HSAs to ensure PSBI identification and follow up. We have updated the discussion to reflect this (TC version, page 25, lines 530-533).

 6. Qualitative interviews - how were these chosen? Were they purposive? Was saturation reached with responses? More detailed methods in this section would make this well-written paper even more robust.

Response: Respondents for qualitative interviews were chosen purposively as described in the methods section. While we did not explicitly define the sample size in terms of saturation, we did find that respondents provided similar answers with regard to our main lines of inquiry, suggesting that data saturation had been reached even with a relatively small sample. We have modified the limitations section (TC version, page 24-25, lines 525-528). 

6. PLOS authors have the option to publish the peer review history of their article (what does this mean?). If published, this will include your full peer review and any attached files. Do you want your identity to be public for this peer review? For information about this choice, including consent withdrawal, please see our Privacy Policy.

Reviewer #1: Yes: Charlotte E Warren

Reviewer #2: Yes: Cyril M Engmann, PATH & University of Washington

---

## [Decision Letter · Decision Letter 1]

4 Feb 2020

Feasibility of implementing the World Health Organization case management guideline for possible serious bacterial infection among young infants in Ntcheu district, Malawi

PONE-D-19-25252R1

Dear Dr. Guenther,

We are pleased to inform you that your manuscript has been judged scientifically suitable for publication and will be formally accepted for publication once it complies with all outstanding technical requirements.

With kind regards,

Emma Sacks

Academic Editor

PLOS ONE

Additional Editor Comments (optional):

Thank you for your thorough revisions. This is an important paper and great contribution to the literature.

Reviewers' comments:

Reviewer's Responses to Questions

**Comments to the Author**

1. If the authors have adequately addressed your comments raised in a previous round of review and you feel that this manuscript is now acceptable for publication, you may indicate that here to bypass the “Comments to the Author” section, enter your conflict of interest statement in the “Confidential to Editor” section, and submit your "Accept" recommendation.

Reviewer #1: All comments have been addressed

2. Is the manuscript technically sound, and do the data support the conclusions?

Reviewer #1: Yes

3. Has the statistical analysis been performed appropriately and rigorously? 

Reviewer #1: Yes

4. Have the authors made all data underlying the findings in their manuscript fully available?

Reviewer #1: Yes

5. Is the manuscript presented in an intelligible fashion and written in standard English?

Reviewer #1: Yes

6. Review Comments to the Author

Reviewer #1: (No Response)

7. PLOS authors have the option to publish the peer review history of their article (what does this mean?). If published, this will include your full peer review and any attached files.

Reviewer #1: Yes: Charlotte E Warren

---

## [Editor Report · Acceptance letter]

2 Mar 2020

PONE-D-19-25252R1 

Feasibility of implementing the World Health Organization case management guideline for possible serious bacterial infection among young infants in Ntcheu district, Malawi 

Dear Dr. Guenther:

I am pleased to inform you that your manuscript has been deemed suitable for publication in PLOS ONE. Congratulations! Your manuscript is now with our production department. 

With kind regards,

on behalf of

Dr. Emma Sacks 

Academic Editor

PLOS ONE